# Numerical Study on the Rising Motion of Bubbles near the Wall

**Kaixin Zhang, Yongzheng Li, Qi Chen and Peifeng Lin** *

Key Laboratory of Fluid Transmission Technology of Zhejiang Province, Zhejiang Sci-Tech University, Hangzhou 310018, China; 202030606392@mails.zstu.edu.cn (K.Z.); 201920501055@mails.zstu.edu.cn (Y.L.); 202130606395@mails.zstu.edu.cn (Q.C.)
* Correspondence: linpf@zstu.edu.cn

**Abstract:** Based on the volume of fluid method (VOF), the rising characteristics of bubbles in near-wall static water are studied. In this study, the influence of the wall on the rising motion of the bubble was studied by changing the distance of the bubble wall, the diameter of the bubble, the arrangement of the bubble and the size ratio, etc. The influence is expressed as the average swing amplitude of the "Z"-shaped motion when the bubble rises. The study found that in the case of a single bubble, the wall surface has a certain influence on the rise of the bubble, and its degree is affected by the bubble wall distance and the bubble diameter. The influence of bubble wall distance is more obvious. The greater the bubble wall distance, the less the bubble is affected by the wall; in the case of double bubbles, the influence of the interaction force between the bubbles is significantly greater than the wall surface.

**Keywords:** gas-liquid two-phase flow; bubbles near the wall; numerical simulation; volume of fluid method





## 1. Introduction

Two-phase flow is a very common phenomenon. As an important branch of fluid mechanics, it began to develop rapidly in the 1960s. It is mainly used to study the interaction between two phases and the changes in flow patterns and motion disciplines. Generally, it is divided into gas-liquid two-phase flow, solid-liquid two-phase flow and liquid-liquid two-phase flow according to the phase state of the substances constituting the flow system [1]. Gas-liquid two-phase flow is one of the most common types of multi-phase flow. It refers to the flow phenomenon of gas-phase fluid and liquid-phase fluid in the same flow system [2,3]. The motion and change of the gas phase in the flow will affect the characteristics of the two-phase flow. Therefore, studying the motion discipline of the gas phase is the key to studying the characteristics of the two-phase flow.

The rise of bubbles in water is a very complicated and unstable process. The bubbles continue to rise in the water. Due to the combined action of gravity, buoyancy and surface tension, bubbles will deform, burst and coalesce. Eventually, the rising path of the bubble will be unstable, usually spiral, "Z"-shaped motion, etc. [4,5]. Bubbles deform and break into many μm-level micro-bubbles during their rising motion. The bubbles usually appear in the form of micro-bubble groups, so it is also very important to study the interaction between bubbles. In this article, the influence of the wall surface on the motion characteristics of single and double bubbles will be explored, and the discipline of motion of bubbles near the wall will be studied.

Since the discipline of bubble motion in liquid is widely used in the chemical industry, bubble dynamics theory was established to study the bubble motion and the interaction between bubbles [6]. At present, many scholars have conducted researches on the motion discipline of a single bubble and the motion discipline of bubble groups. The American scientist Clift et al. [7,8] studied the shape change of bubbles in the liquid after consulting a large number of data in the literature. According to the dimensionless criterion, they drew

the bubble shape phase diagram, which is called the Grace diagram. Some scholars [9–11] also verified the bubble shape phase diagram obtained by Clift, and found that the result is consistent with the bubble phase diagram. Duineveld [12] found that when the equivalent diameter of the bubble is less than 1.8 mm, because the bubble shape tends to be spherical, the rising path presents a straight line; when the equivalent diameter of the bubble exceeds 1.8 mm, the path of bubble will change from a straight line to a "Z" shape. The bubble-free surface interaction has been experimentally investigated by Robinson et al. [13], when bubbles are close together, their mutual interactions can influence the existence of free surfaces. These factors can influence the migratory behavior and lifetime of the bubbles. Sugiyam et al. [14] studied the near-wall motion of deformable bubbles in viscous fluids, and found that the bubbles have a significant lateral deviation in the wall domain. When the distance between the bubble and the wall changes, the bubble will bounce due to the pressure difference between the two sides and the lift between the wall, which is the "Z"-shaped motion path mentioned above. Chen et al. [15] tracked the three-dimensional deformation of the two-phase interface based on the numerical simulation method of Front Tracking and compared the data with unbounded still water. They found that the bubbles around the wall would be suppressed by the wall, causing the bubbles to gradually deviate from the wall. Tatineni et al. [16]further proposed a high-resolution numerical method based on the level set method to solve the motion and deformation of the gas-liquid interface. It is found that the rising bubbles will be affected by the wall surface and migrate laterally to the central domain when they are close to the wall surface. And the closer the bubble wall is, the greater the influence of the wall on the bubble. Many scholars have also conducted research on the rise of multiple bubbles. Komasawa et al. [17] studied the motion characteristics of stationary bubbles in downward flowing water, and found that the experimental results are consistent with the motion characteristics of bubbles rising freely. At the same time, they also studied the effect of bubble wake on bubbles. Ohta et al. [18] examined the influence of initial bubble conditions on bubble rise motion, two-dimensional direct numerical simulations of the motion of a gas bubble rising in viscous liquids were carried out by a coupled level set/volume-of-fluid method. The results show that the smaller the distance between two bubbles, the greater the influence on their respective velocity fields, and the easier it is for the bubbles to attract and fuse with each other.

In this article, the motion characteristics of bubbles in water are studied by using the VOF method in Fluent. By changing the bubble diameter, bubble wall distance, etc., the motion path, rising speed, and surrounding flow field changes of the bubbles in the near-wall domain are studied to analyze the changes in the bubble motion characteristics. On the basis of the single-bubble research, the double-bubble research is carried out. The double-bubble is used to analyze the interaction between the bubbles, and the bubble interaction and wall influence are analyzed and studied by changing the arrangement of the bubbles.

## 2. Numerical Simulation Method

### 2.1. Governing Equation

The problem of two-phase interface is usually calculated by capturing the phase interface and performing numerical simulation. The more commonly used methods are the Volume of Fluid (VOF) method [19–21] and the Level Set (LS) method [22,23]. In this article, the VOF method is used to track the gas-liquid interface to study the motion of bubbles. The VOF method establishes the two-phase intersection interface based on the volume ratio function. The sum of the volume fraction of each phase is 1, so the maximum value is 1 [24]. Since the phases are not interspersed with each other, each additional phase only needs to increase the volume fraction function of the corresponding phase. In the VOF model, suppose there is a two-phase fluid, and mark the calculation domain as $S$, then the domain where fluid $A$ and fluid $B$ are located can be represented by $S^A$ and $S^B$.

For the continuity equation of incompressible fluid, the governing equation of bubble motion is:

$$\nabla \cdot u = 0 \tag{1}$$

$$\frac{\partial \alpha_A}{\partial t} + u_A \nabla \alpha_A = \frac{S^A}{\rho_A} + \frac{1}{\rho_A} \sum_{B=1}^{n} = 1(\dot{m}_{BA} - \dot{m}_{AB}) \tag{2}$$

where $\alpha_A$ is the volume fraction of the fluid A; $u_A$ is the velocity vector of the fluid A, m·s$^{-1}$; $\rho$ is the density, kg·m$^{-3}$; $m_{BA}$ is the mass transport from the B (A) phase to the A(B) phase, kg.

Where $\overline{V}(u,v)$ is the velocity field of the fluid, $\Delta V_{ij}$ is the volume of a single grid, and $Q_{ij}$ is defined on each grid $I_{ij}$ as the integral of $\alpha(\overline{x}, t)$ on the grid:

$$Q_{ij} = \frac{1}{\Delta V_{ij}} \int_{I_{ij}} \alpha(\overline{x}, t) dV \tag{3}$$

Equation (3) is the VOF function, and it also satisfies:

$$\frac{\partial Q}{\partial t} + u \frac{\partial Q}{\partial x} + v \frac{\partial Q}{\partial y} = 0 \tag{4}$$

Equation (4) is the VOF equation. From the above, it can be seen that the volume function of each phase is essentially the ratio of the phase volume to the grid volume in the grid.

The momentum conservation equation considering surface tension is as follow:

$$\rho \left[ \frac{\partial u}{\partial t} + (u \cdot \nabla)u \right] = -\nabla p + \nabla \cdot (2\mu D) + \rho g + \varphi \tag{5}$$

where $\varphi$ is surface tension, $p$ is pressure, $\mu$ is dynamic viscosity coefficient.

*2.2. Geometric Model*

In the calculation of the two-dimensional plane domain, in order to ensure that the bubble has obvious deformation, and will not cause the bubble rising instability due to excessive deformation, the initial bubble with a diameter of $d = 4$ mm is selected and simulated. The entire domain is 80 mm long and 160 mm high. The height of the liquid level is 140 mm and the initial height of the bubble is 20 mm. The distance from the bubble to the bottom surface is 5 times its diameter, and the distance from the non-researched side wall is greater than 10 times the diameter, which can eliminate the influence of the non-researched side wall and the bottom surface of the bubble motion as much as possible. Taking a bubble with an initial diameter of 4 mm and a bubble wall distance of 4 mm as an example, the calculation domain is shown in Figure 1. Other bubbles with different diameters are the same as in this case.

The mesh of the model is drawn by ICEM software. The two groups of models in this article have regular structures and simple shapes, and the mesh is divided into structured grids.

Figure 2 is the grid division of the two-dimensional calculation model. The quadrilateral structure grid is used. The minimum size of the grid is 0.2 mm. Due to the need to study the motion of the wall bubbles, boundary refinement is used, and the overall grid number is 150,000. The gas used in the simulation is air, and the liquid is liquid water. Both physical properties and parameters under normal temperature and pressure are used. Except for the verification of the VOF method, the same gravity and surface tension are used. The surface tension of liquid is 0.0728 N/m.

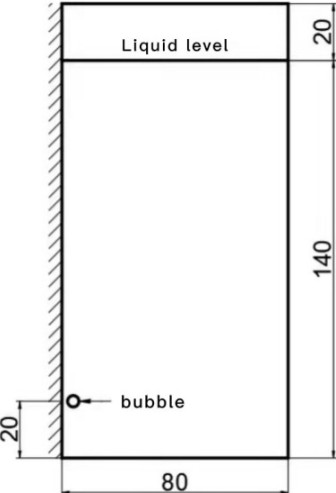

**Figure 1.** Two-dimensional calculation domain of the near-wall bubble with an initial diameter of 4 mm.

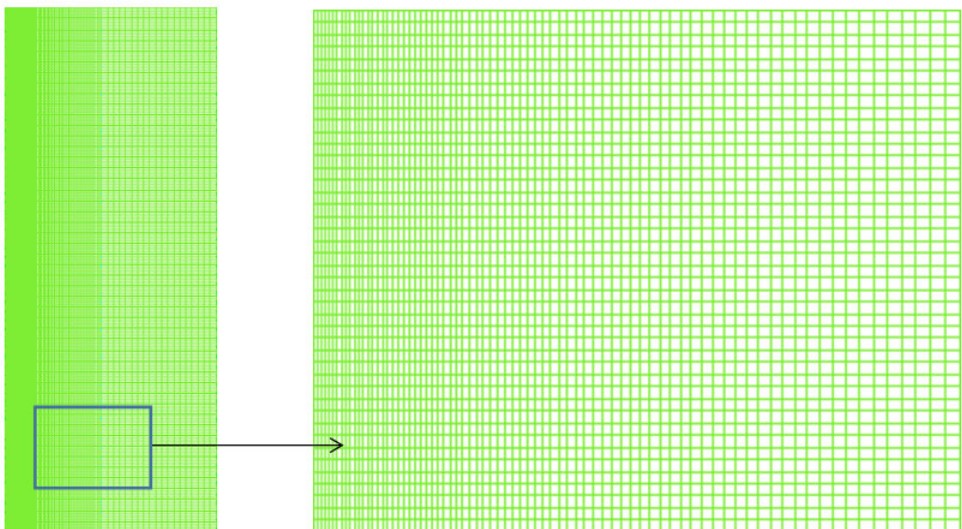

**Figure 2.** Global grid and local grid of 2D model.

In the two-dimensional model, the top is set to PRESSURE-OUTLET, and the gauge pressure is 0 Pa; the rest of the walls are set to WALL, and the wall uses no-slip boundary conditions.

### 2.3. Grid Independence and Algorithm Verification

For the research on the rise of bubbles with a diameter of 4 mm in the two-dimensional calculation domain, this article divides into four kinds of grids of 80,000, 100,000, 150,000 and 200,000 to verify the independence of the grid. This research refers to the verification method for grid independence in the article by Wang [25]. The position change of the top and bottom of the bubble over time was used to verify the grid independence. The top and bottom positions of the bubble are shown in Figure 3.

Figure 4a,b are the curves of the height of the top and bottom of the bubble rising with time. It can be seen from the curve that the number of grids in the four cases basically coincides with the curve, and there is no big difference. Therefore, the grid independence is verified. In this study, 150,000 grids are used for numerical simulation.

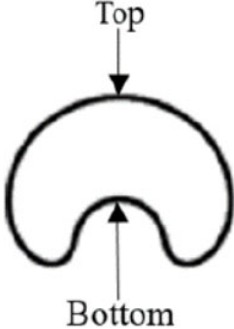

**Figure 3.** Schematic diagram of the top and bottom of the bubble in the literature.

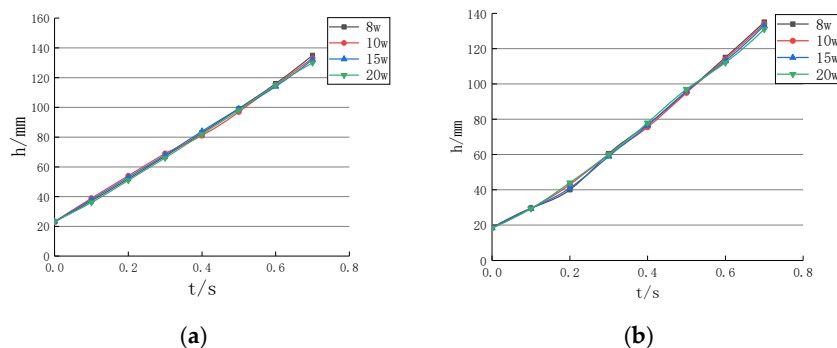

(**a**)                  (**b**)

**Figure 4.** The change curve of bubble position with time under different grid numbers: (**a**) Top (**b**) Bottom.

Before the numerical simulation, the correctness of choosing the VOF model for calculation is first verified in this article. In Figure 5, Wang et al. [25] studied the rising deformation of bubbles with different diameters. In our current project, the bubble diameters are all below 10 mm, the rising deformation of 7.5 mm bubbles is verified in Figure 6, so it is proved that this article chooses to use VOF for calculation.

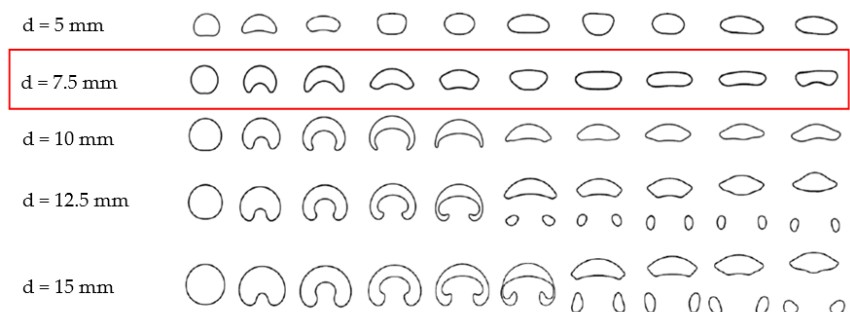

**Figure 5.** Deformation during rising of bubbles of different diameters.

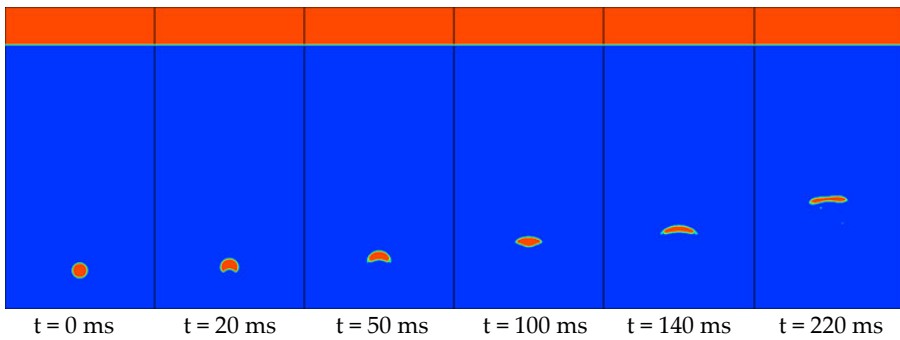

t = 0 ms      t = 20 ms      t = 50 ms      t = 100 ms      t = 140 ms      t = 220 ms

**Figure 6.** The rising and deformation process of a single central circular bubble without boundary.

## 3. Numerical Simulation Results

### 3.1. Analysis of Single Bubble Motion near the Wall

When the bubble rises freely, the bubble will rise in a straight line; when the bubble is located near the wall, the bubble will swing and rise in a "Z" shape. In this section, when the bubble diameter is fixed at 4 mm, the bubble wall distance and bubble diameter are changed, and the influence of the wall surface on the rise of the bubble is analyzed. In this article, the distance is defined as the dimensionless quantity $S^*$.

$$S^* = \frac{D_e}{D_0} \tag{6}$$

where $D_e$ is the distance between the wall and the center of the bubble, that is, the distance from the bubble wall, $D_0$ is bubble diameter, $L_n$ is the last observable swing amplitude of the bubble, $T_n$ is the last observable full cycle of the bubble swing.

3.1.1. The Influence of Bubble Wall Distance

Figure 7a is the ascending path of the 4 mm bubble at $S^* = 0.75$, 1.0, 1.25, 1.5. The bubble rises linearly in a certain path, and then begins to deviate to the side away from the wall to swing up in a "Z" shape. Figure 7b adjusts the origin of the X axis of the four types of bubbles. The origin of the X axis is set to the X coordinate corresponding to the center of the bubble, so that the starting points of the bubble path are all fitted together. The path difference between the bubbles can be seen more intuitively.

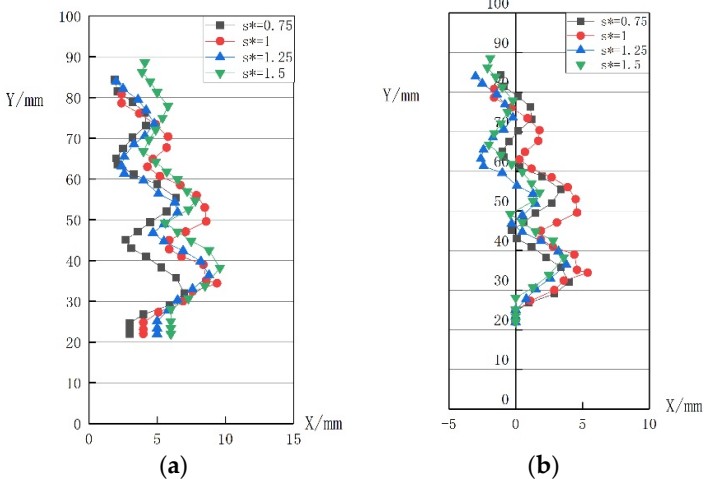

**Figure 7.** Bubble motion path diagram under different S*: (**a**) Different initial position, (**b**) Same initial position.

In the Table 1, $L$ is the average swing amplitude of the bubble, and $F$ is the average swing frequency of the bubble. When $S^*$ becomes larger, the bubble gradually moves away from the wall, and the swing amplitude of the bubble gradually decreases, while the swing frequency of the bubble increases. The influence of the wall on the bubble gradually weakens as the bubble moves away. In the table, $H_{max}$ represents the maximum rise height of the bubble during the 500 ms monitoring time. In $H_{max}$, the rise height is the lowest when $S^* = 1$, and when $S^*$ becomes larger, the bubble height also becomes higher.

**Table 1.** Numerical table of bubble rise swing under different S*.

| S* | $L_1$/mm | $L_2$/mm | $L_3$/mm | $L_4$/mm | $L_5$/mm | L/mm | $H_{max}$/mm | F/s$^{-1}$ |
|---|---|---|---|---|---|---|---|---|
| 0.75 | 4.0 | 4.3 | 3.4 | 4.4 | 2.2 | 3.66 | 84.4 | 4.16 |
| 1 | 5.4 | 3.5 | 2.7 | 4.3 | 1.5 | 3.48 | 80.8 | 4.42 |
| 1.25 | 3.8 | 4.1 | 1.8 | 4.1 | 2.4 | 3.22 | 84 | 4.69 |
| 1.5 | 3.6 | 4.0 | 2.2 | 3.8 | 1.8 | 3.08 | 88.6 | 4.85 |

Figure 8 shows the average rising speed of the bubble every 20 ms in 500 ms. From Table 2, it can be seen that when the bubble rises, the speed will gradually increase, and when it reaches a certain value, it will gradually stabilize and fluctuate up and down within a certain range. The farther the bubble is from the wall, the faster the initial rising speed increases. From the two curves of $S^* = 1$ and $S^* = 0.75$, it can be found that when the bubble rises in contact with the wall, the speed will be faster than the free rise of the bubble. It can be seen from Table 2 that the overall rising speed of the bubble within 500 ms can also prove that the bubble bounce motion will rise faster than the free motion.

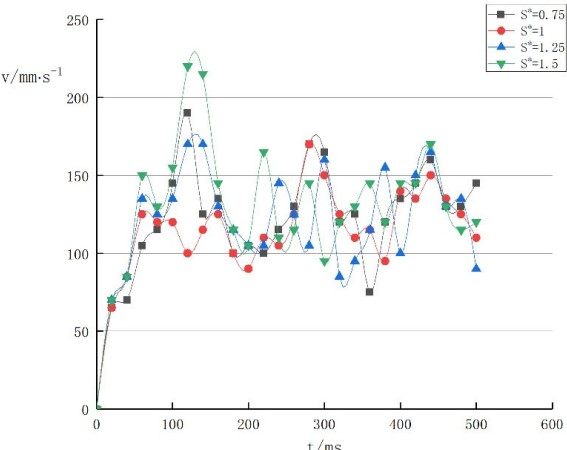

**Figure 8.** Graphs of the average rising velocity of bubbles under different S*.

**Table 2.** Numerical table of rising swing amplitude of bubbles with different diameters.

| Bubble Diameter /mm | $L_1$/mm | $L_2$/mm | $L_3$/mm | $L_4$/mm | $L_5$/mm | L/mm | $H_{max}$/mm | F/s$^{-1}$ |
|---|---|---|---|---|---|---|---|---|
| 4 | 3.6 | 4.0 | 2.2 | 3.8 | 1.8 | 3.08 | 88.6 | 4.85 |
| 5 | 3.7 | 5 | 4.2 | 3.1 | – | 4.0 | 94.4 | 4.24 |
| 6 | 3.4 | 3.7 | 7.3 | 4.6 | – | 4.75 | 99 | 3.57 |

It can be seen from Figure 9a,b that the flow field of bubbles rising near the wall is different from the flow field of bubbles rising in unbounded still water. Due to the influence of the left wall, the flow field on the left side of the fluid flows closer to the left side of the bubble, and the flow velocity direction is more. It is the tangent direction of the left side of the bubble. The flow velocity on the left side will be faster than that of unbounded static water, resulting in the formation of the flow field at the bottom of the bubble forming a situation where the left side is less and the right side is more, which further causes the clockwise deflection of the bubble.

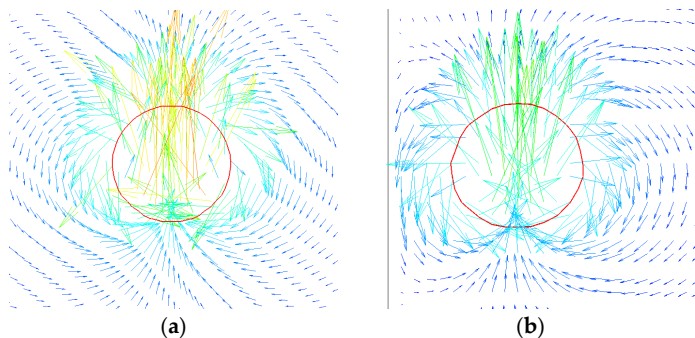

**Figure 9.** The velocity vector generated by the near-wall bubble and the central bubble rising 10 ms: (**a**) The central bubble, (**b**) The near-wall bubble.

### 3.1.2. Influence of Bubble Diameter

Table 2 shows the swing amplitude when the bubble rises with diameters of 4, 5, and 6 mm and the rise height within 500 ms when $S^* = 1.5$. It can be seen from the table that when the bubble diameter becomes larger, the average swing amplitude of the bubble increases.

Figure 10 during the rising process of the bubble, the overall upward path is still a straight upward first, and then a "Z"-shaped swing upward. However, it can be found that as the diameter increases, each deflection point of the bubble is rising, which means that the bubble swing frequency is decreasing, the bubble swing amplitude is increasing, and the average rising speed within 500 ms is also increasing. But the difference from the previous section is that the initial bubble rise speed decreases as the bubble diameter increases. It can also be seen in the figure that as the diameter of the bubble increases, the overall upward path of the bubble gradually deviates from the wall surface.

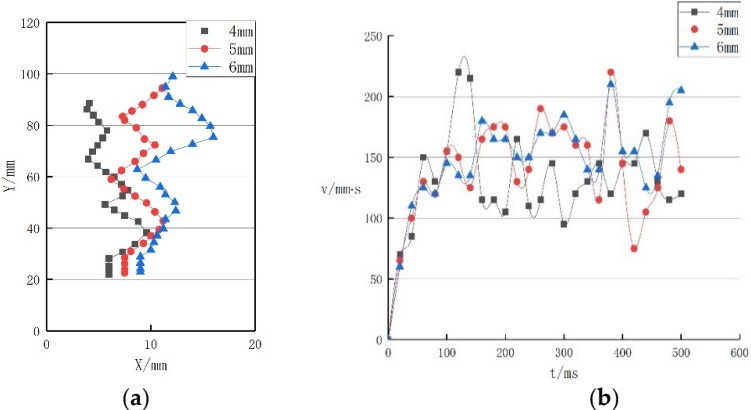

(**a**)  (**b**)

**Figure 10.** Bubble motion path and rising speed diagram under different diameters: (**a**) Motion path, (**b**) Ascent speed.

### 3.2. Analysis of the Motion of Double Bubbles near the Wall

3.2.1. Vertically Rising Bubbles

(1)  Double bubbles of equal diameter

Figure 11 shows the upward motion of two vertically arranged bubbles in the near-wall domain. The diameter of the bubbles is 4 mm, and the two bubbles are arranged vertically with a distance of 8 mm. At 100 ms, it can be found that the two bubbles are deflected away from the wall under the influence of the wall; At 115 ms, the deflection of the tail bubble recovers and it approaches the wall again, while the head bubble continues to move away from the wall; A large bubble is formed after two bubbles come into contact at 140 ms. From the bubble motion in the figure and the previous research, we can find that two bubbles with the same diameter and the same distance from the bubble wall produce different swing amplitudes in the near-wall domain. This shows that the effect of the near wall on the double bubble is different from that of the single bubble.

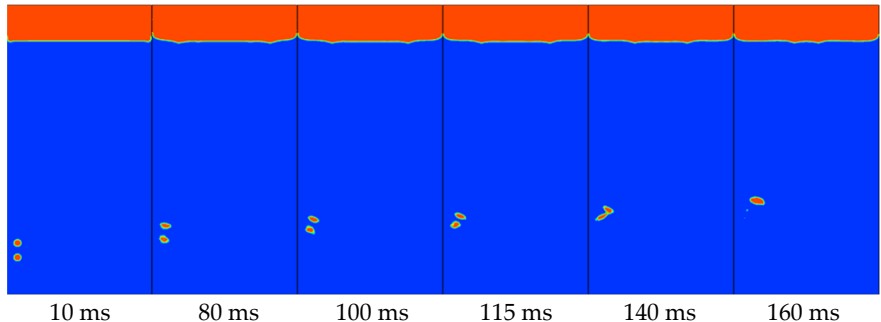

10 ms  80 ms  100 ms  115 ms  140 ms  160 ms

**Figure 11.** The shape change diagram of the rising double bubbles vertically arranged near the wall.

Figure 12 is the flow field diagram of the bubble at different moments. At 80 ms, the bubble moves away from the wall due to the influence of the wall at this time. At 110 ms, the bubble forms a wake vortex due to the deflection, and the right wake vortex of the head bubble has an impact on the tail bubble, causing the bubble to rotate counterclockwise to restore the deflection, resulting in a difference in the swing amplitude of the two bubbles.

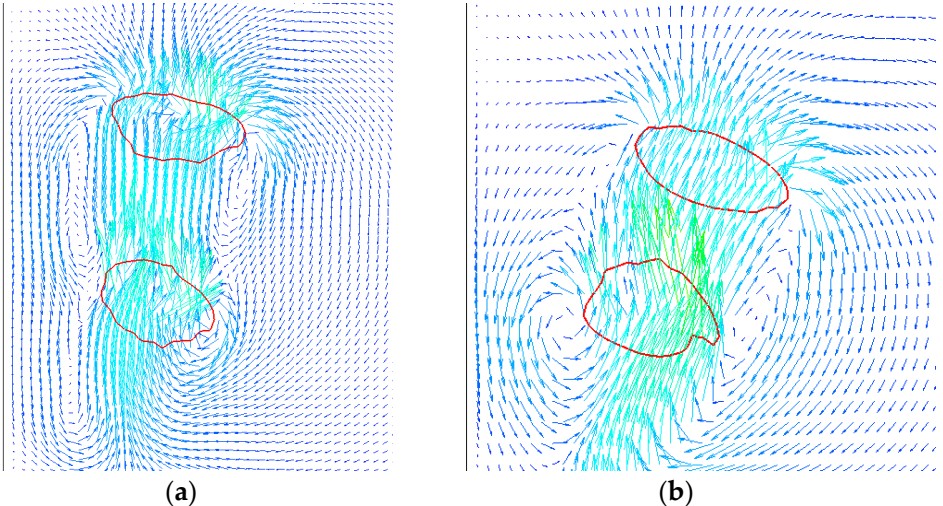

(**a**)  (**b**)

**Figure 12.** Upward flow field diagrams of vertically arranged double bubbles at different moments: (**a**) 80 ms, (**b**) 110 ms.

(2)  The upper small and the lower large double bubbles

Figure 13 is a simulation of the rising motion of double bubbles of different sizes arranged vertically, with small head bubbles and large tail bubbles. From the figure, it can be found that the double bubbles of the same size are similar to those in the previous article. Both bubbles deflected away from the wall, and then merged.

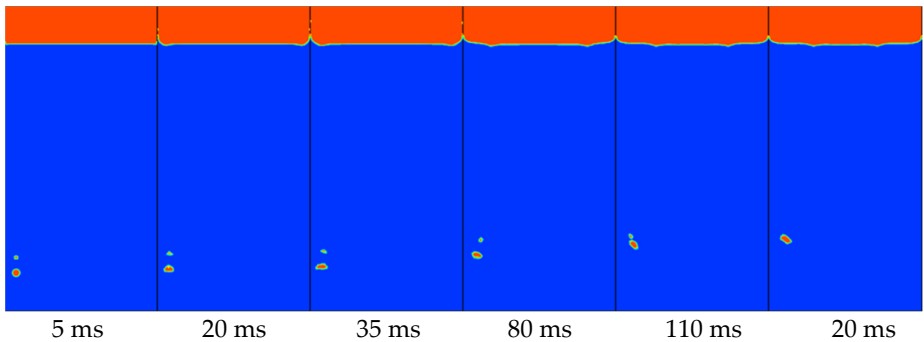

5 ms  20 ms  35 ms  80 ms  110 ms  20 ms

**Figure 13.** The rising shape of double bubbles of different sizes near the wall (upper small and lower large).

In Figure 14a, it can be found that the flow field of large bubbles is denser than that of small bubbles. This is the reason why the rising speed of large bubbles is faster than that of small bubbles; the big bubble is moving away from the wall, while the small bubble has recovered its deflection and is moving close to the wall. This result can be mutually confirmed with the near-wall motion of single bubbles with different diameters. Large bubbles are affected by the wall surface more obviously, with larger average swing amplitude and lower frequency.

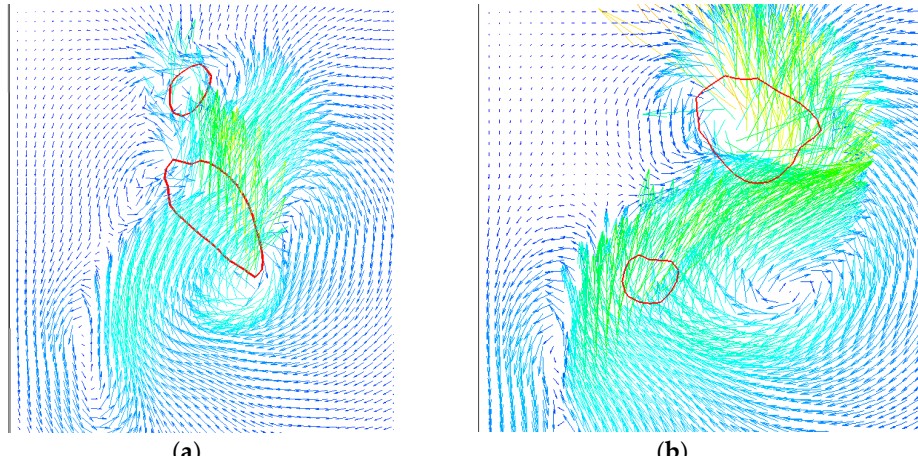

**Figure 14.** Upward flow field diagram of double bubbles of different sizes near the wall at 110 ms: (**a**) Upper small and lower large, (**b**) Upper big and lower small.

(3)     The upper large and the lower small double bubbles

Figure 15 is a simulation of the rising motion of double bubbles of different sizes arranged vertically. The head bubble is large and the tail bubble is small. Contrary to the previous section, the bubbles still coalesce. It can be seen from the figure that it is different from the previous section. The big bubble first deflects at 70 ms, while the small bubble starts to deflect for the first time at 135 ms. The speed of the small bubble is faster than that of the big bubble. The two bubbles come into contact at 160 ms and then merge into one large bubble.

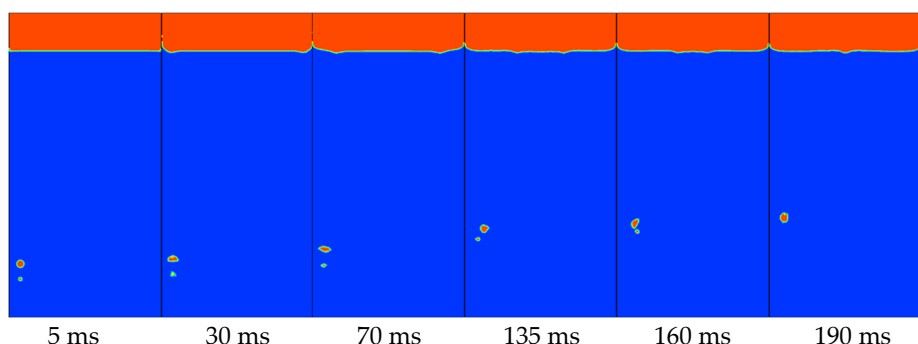

|  5 ms | 30 ms | 70 ms | 135 ms | 160 ms | 190 ms |

**Figure 15.** The rising shape of double bubbles of different sizes near the wall (upper large and lower small).

From Figure 14b, it can be seen that when the big bubble rises, a wake vortex is left at the tail. The small bubbles rise rapidly along the jet formed by the rising of the large bubbles, thus coming into contact with the large bubbles.

### 3.2.2. Horizontally Arranged Rising Bubbles

(1)     Horizontal arrangement with 2 times diameter spacing

Figure 16 is a simulation of the upward motion of double bubbles arranged horizontally. The motion of the double bubble near the wall is similar to the motion of the free double bubble. The two bubbles separated first, then approached, and finally merged together. Compared with a single bubble, it can be found that the left bubble is not affected by the wall surface, and first deviates from the wall surface.

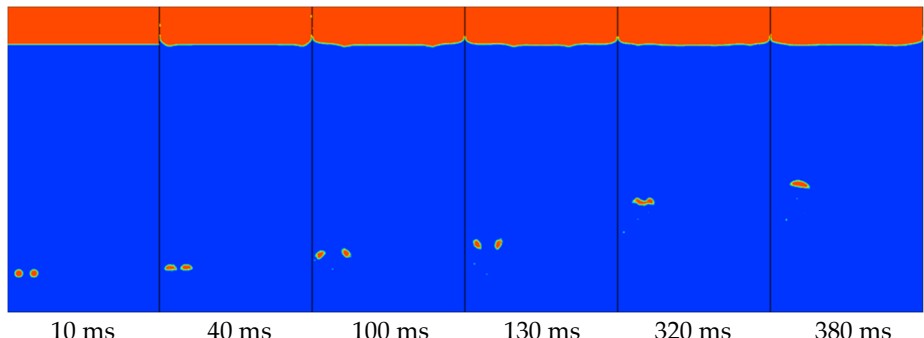

| 10 ms | 40 ms | 100 ms | 130 ms | 320 ms | 380 ms |

**Figure 16.** The shape change of the rising double bubbles arranged vertically near the wall.

Figure 17 shows the surrounding flow field distribution of two bubbles at 50 ms. At this time, the bubbles are beginning to move away from each other. When there is a single bubble near the wall, due to the influence of the left wall, the flow field on the left side of the fluid will flow closer to the left side of the bubble. The direction of the flow velocity is more the tangent direction of the left side of the bubble, and the flow velocity on the left will be faster than that of unbounded still water. As a result, the flow field at the bottom of the bubble forms a situation where the left side is less and the right side is more. In the case of double bubbles, it can be seen from the figure that two jets merge between the two bubbles, causing the two bubbles to have a fast flow velocity in the middle of the bubble and a slow flow velocity on both sides, thus forming a deflection of the two bubbles away from each other. The flow field on the left side of the left bubble is affected by the wall, and the flow field on the right side of the right bubble has no boundary. The bubble on the left will swing wider than the bubble on the right, and the overall path of the two bubbles will gradually deviate from the wall.

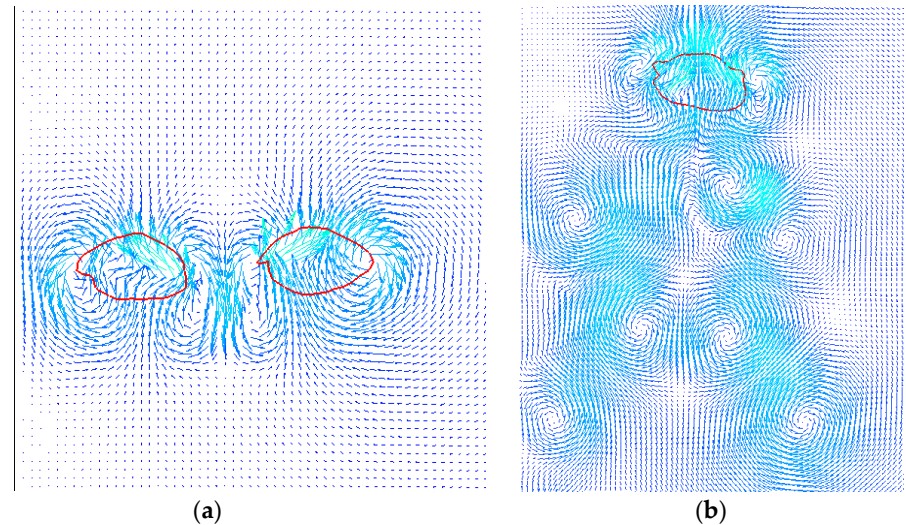

(**a**)      (**b**)

**Figure 17.** The flow field distribution diagram of the horizontally arranged near-wall double bubbles at the initial stage of rise and after the coalescence: (**a**) Bubbles before merging, (**b**) The merged bubbles.

(2)    Double bubbles arranged horizontally at 3 times diameter spacing

Figure 18 is the rising deformation diagram of the horizontally arranged double bubbles with the bubble spacing of 3 times the bubble diameter. It can be seen from the figure that the two bubbles did not coalesce, and the interaction between the bubbles became smaller due to the increase in the distance between the bubbles. So unlike the previous article, the bubbles are far away from each other, The left bubble also appeared to move away from the wall at 90 ms, and the left bubble came into contact with the wall

at 500 ms. The rising form changes to bounce rising, and it is obvious that the left bubble rises faster at 850 ms. It is mutually corroborated with the conclusion of the bubble rising speed in the previous article.

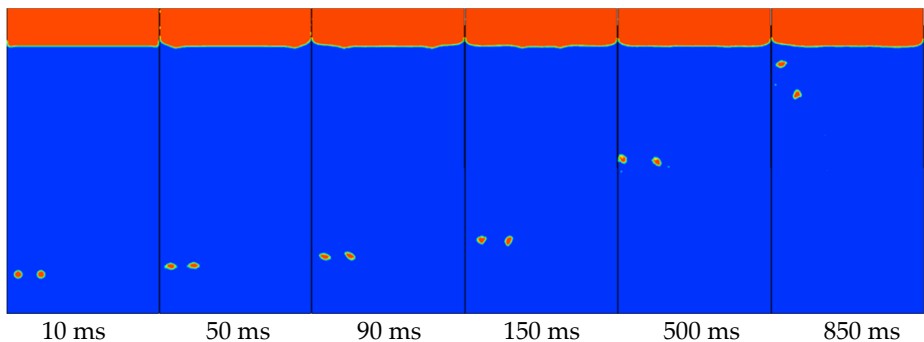

10 ms    50 ms    90 ms    150 ms    500 ms    850 ms

**Figure 18.** The shape change of the rising double bubbles arranged horizontally near the wall.

Figure 19 shows the flow field distribution of the bubble at 90 ms. It can be found that the deflection of the left bubble is slightly smaller than that of the right bubble. This is because the inducement of the deflection of the left bubble is that the influence of the wall surface is more obvious than the influence between the bubbles, and the jet formed on the wall promotes the deflection of the left bubble; The right bubble is the bubble deflection caused by the interaction between the bubbles, so compared with the two, the right bubble deflection is greater.

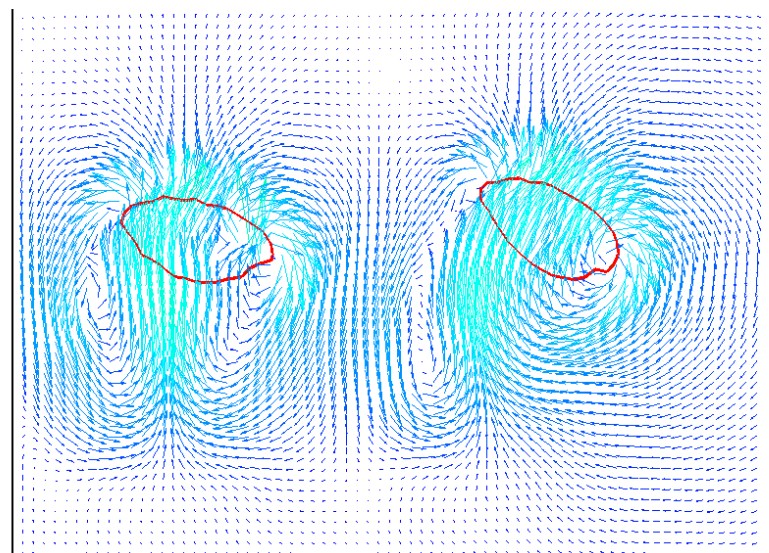

**Figure 19.** The upward flow field of double bubbles arranged horizontally near the wall at 90 ms.

(3)　Double bubbles arranged horizontally at 4 times diameter spacing

Figure 20 shows the rising deformation of horizontally arranged double bubbles with a bubble spacing of four times the diameter. It can be seen from the figure that at four times the diameter, the interaction between the bubbles becomes very insignificant, the right bubble basically rises in a straight line, and the rising speed of the right bubble is significantly faster than the left bubble at 850 ms. It can be confirmed that when the distance between the bubble wall and the bubble wall gradually increases, the influence of the wall surface on the bubble will gradually weaken, and the rising speed of the bubble will gradually increase.

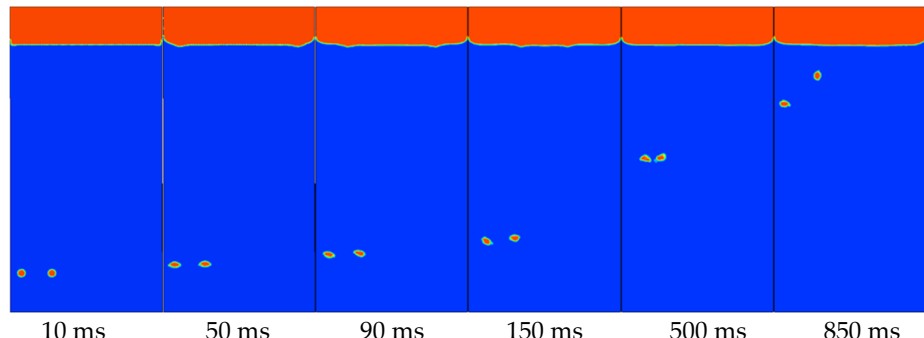

**Figure 20.** The shape change of the rising double bubbles arranged horizontally near the wall.

In summary, when the vertically arranged double bubbles of different sizes rise, with proper bubble spacing, coalescence can occur regardless of whether it is an arrangement of large top and small bottom or large top and small bottom. When the head bubbles are small, the reason for coalescence is that the big bubbles rise faster than the small bubbles; when the head bubbles are large, the coalescence is due to the jet flow generated when the large bubbles rise to accelerate the rising speed of the small bubbles.

## 4. Conclusions

This article takes the rising motion of bubbles in the near-wall domain as the research topic. Numerical simulations are used to analyze the rising motion characteristics of a single circular bubble and double circular bubbles in the domain near the wall, and explore the influence of the wall on the bubble rising, and the following conclusions are obtained:

(1) When a single bubble rises near the wall, the bubble will be affected by the wall and change its original state of motion. The bubble first rises in a straight line, and then begins to swing upward. When $S^*$ becomes larger, the height of bubble rising also becomes higher.

(2) Under the condition that the diameter of the bubble is fixed, when $S^* < 1$, the bubble will touch the wall when it rises, and then form a bouncing upward motion; and when $S^* \geq 1$, the bubble will not touch the wall, and the bubble will rise straight up for a certain distance and then rise in a "Z" shape.

(3) Numerical simulation results of the rising motion of a single bubble with a fixed diameter show that when $S^*$ increases from 0.75 to 1.5, the bubble's average swing amplitude decreases from 3.66 mm to 3.08 mm, while the bubble's swing frequency decreases from 4.16 s$^{-1}$ increases to 4.85 s$^{-1}$. The results of this study show that the distance of the bubble wall has an effect on the rising movement of the bubble. It is concluded that when the diameter is fixed, when $S^*$ gradually becomes larger, that is, when the bubble gradually moves away from the wall, the swing amplitude of the bubble gradually decreases, and the swing frequency of the bubble follows. As the bubble increases, the influence of the wall on the bubble gradually weakens as the bubble moves away.

(4) The result of numerical simulation of the rising motion of a single bubble with $S^* < 1$ shows that when the diameters are 4, 5, and 6 mm, the average swing amplitude of the bubble increases from 3.08 mm to 4.75 mm, and the swing frequency of the bubble increases from 4.85 s$^{-1}$.The law of change reduced to 3.57 s$^{-1}$. The results of this study indicate that the bubble diameter has an effect on the bubble's rising motion. It is concluded that when the bubble diameter gradually increases in the area near the wall, the swing amplitude of the bubble gradually increases, while the swing frequency of the bubble decreases accordingly. It is concluded that the influence of the size of the bubble on the rising movement of the bubble gradually increases with the larger the bubble diameter.

(5) The double bubbles in the near-wall domain will be affected by the wall and move away from the wall. The swing amplitude of the head bubble is greater than that of the tail bubble.

(6) When the double bubbles are arranged horizontally, the influence of the wall surface on the bubbles is not obvious. The bubbles first move away from each other and then approach, and the flow field after the bubbles merge is symmetrically distributed. When the distance between the horizontally arranged double bubbles gradually increases, the influence of the wall surface on the left bubble will gradually become obvious. When the vertically arranged double bubbles of different sizes rise, with proper bubble spacing, coalescence can occur regardless of whether it is an arrangement of large top and small bottom or large top and small bottom.

**Author Contributions:** Conceptualization, P.L.; Formal analysis, Y.L. and K.Z.; Investigation, Q.C. and P.L.; writing-original draft preparation, K.Z.; writing-review and editing, Y.L. All authors have read and agreed to the published version of the manuscript.

**Funding:** The present work is financially supported by the Key R&D Program of Zhejiang Province (Grant No. 2020C03081), the Joint Funds of the National Natural Science Foundation of China (Grant No. U2006221), the National Natural Science Foundation of China (Grant No. 51676173).

**Conflicts of Interest:** The authors declare no conflict of interest.

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
