# Peer review of "Numerical Study on the Rising Motion of Bubbles near the Wall"

_applsci, doi:10.3390/app112210918_

Round 1

Reviewer 1 Report

This is a nice numerical study worthy of a conference paper but lack of validation and simplified modeling limits its utility for archival publication. For example, the choice of dimensions is arbitrary. There is little reference to relevant applications or exploration of theory.

Why did the study use 2-D model - why not 3-D? 

Suggest presenting majority of results in dimensionless form, e.g., normalize domain dimensions and wall position by bubble diameter.

Minor point - overuse of symbol F for several parameters - confusing.

Reviewer 2 Report

The authors have presented the rising properties of bubbles in near-wall static water based on the volume of the fluid method (VOF). The influence of the wall on the rising motion of the bubble was tested out by changing the most important control factors as the distance of the bubble wall, the diameter of the bubble, the arrangement of the bubble, and the size ratio, etc. The greater the bubble wall distance, the less the bubble is affected by the wall.

In my opinion, the paper is interesting, especially in the novel topic range.

Strengths

The greatest advantage of the paper, in my opinion, is the modern subject and the use of the novell Computational Fluid Dynamic (CFD) technique in research.

Weaknesses

1) The biggest weakness of this paper is extremely sloppy formatting, especially glaring in terms of references to literature. In such a state, the work cannot even be read with understanding, and there is no question of an objective review. 

2) The references to literature are quite old. There are no items from the last 2 years, there is only 1 item from the last 3 years. It would be a bit refreshing.

Small errors do not diminish the value of work, but they must absolutely be improved:

Applies to all work: we put a space between the value and the unit for example: 

Line 125, is: 80mm; should be: 80 mm

Table1: The numeric values in the table are messily formatted. This makes reading difficult.

Line 43,        is: Clift et al.[5 ,6];                       should be:  …Clift et al. [5,6]

Lines 51-52, is: Hira-ta[11]injected;                should be: Hira-ta [11] injected

Line 56,        is: Sugiyam et al.[12]studied ;     should be: Sugiyam et al. [12] studied  

Lines 60-61, is: Chen et al.[13]tracked;            should be: Chen et  al. [13] tracked  

Line 64,        is: Tatineni et al.[14]further ; should be: Tatineni et al. [14] further  

Line 68,          is: surface .And  ;                 should be: surface. And  

Line 95,          is: 1[19]. ;                             should be: 1 [19].  

Lines 131, 138, 160, 162, 194, 196, 212, 213, 218, 221, 235, 239, 259, 274, 281, 294, 301, 318, 327, 349, 360, 368,  Error! Reference source not found..

Line 185 what is meaning „In,”?

References:

2. Capital letters are redundant.

18. Journal of Huazhong University of ence and Technology - New journal or new University???

Round 2

Reviewer 1 Report

Changes and explanations are acceptabe.